# Light control of the peptide-loading complex synchronizes antigen translocation and MHC I trafficking

Jamina Brunnberg[1,2], Valentina Herbring[1,2], Esteban Günther Castillo [1], Heike Krüger[1], Ralph Wieneke [1] & Robert Tampé [1✉]

Antigen presentation via major histocompatibility complex class I (MHC I) molecules is essential to mount an adaptive immune response against pathogens and cancerous cells. To this end, the transporter associated with antigen processing (TAP) delivers snippets of the cellular proteome, resulting from proteasomal degradation, into the ER lumen. After peptide loading and editing by the peptide-loading complex (PLC), stable peptide-MHC I complexes are released for cell surface presentation. Since the process of MHC I trafficking is poorly defined, we established an approach to control antigen presentation by introduction of a photo-caged amino acid in the catalytic ATP-binding site of TAP. By optical control, we initiate TAP-dependent antigen translocation, thus providing new insights into TAP function within the PLC and MHC I trafficking in living cells. Moreover, this versatile approach has the potential to be applied in the study of other cellular pathways controlled by P-loop ATP/GTPases.

[1] Institute of Biochemistry, Biocenter, Goethe University Frankfurt, Frankfurt am Main, Germany. [2]These authors contributed equally: Jamina Brunnberg, Valentina Herbring. ✉email: tampe@em.uni-frankfurt.de

Antigen presentation via major histocompatibility complex class I (MHC I) molecules is one of the most elaborate defense strategies of the adaptive immune system against pathogens and tumor cells[1,2]. A key player of this antigen presentation pathway is the heterodimeric transporter associated with antigen processing TAP, which belongs to the superfamily of ATP-binding cassette (ABC) proteins[3]. Each TAP subunit (TAP1/2, ABCB2/B3) can be divided into its core transporter region that comprises six transmembrane helices linked to a nucleotide-binding domain and the TMD0, an extra transmembrane domain with four helices N-terminally attached to each of the core subunits[4]. Notably, TAP is part of the peptide-loading complex (PLC) located in the ER membrane and operates as a gatekeeper for the pool of antigenic peptides derived from cytosolic proteasomal degradation[1,2]. To ensure MHC I antigen presentation, TAP transports peptides that have an overlapping length and sequence specificity for the interaction with MHC I molecules[5,6]. In the ER lumen, longer peptides can be processed by the aminopeptidases ERAP1/2 to match with the MHC I peptide-binding pocket[7–9]. The chaperones calreticulin[10,11] and ERp57[12,13] assist in peptide loading by ensuring the correct assembly of MHC I molecules and the editing module, while tapasin is involved in peptide proofreading[14,15]. The architecture of the native PLC illustrates the molecular synergy of antigen translocation and ER quality control, including specific MHC I chaperoning and editing[16]. Optimally loaded peptide-MHC I complexes dissociate from the PLC and travel via the secretory pathway to the cell surface to elicit a CD8[+] T-cell response. Although MHC I presentation is well studied, our knowledge on the trafficking of antigenic peptides and rate-limiting processes is still limited. Typically, antigen processing has been explored by gene silencing, gene editing, and gene knockouts, which are limited by compensation and feedback effects. Hence, new approaches to spatiotemporally control key steps in the MHC I surface presentation pathway are eagerly awaited.

Genetic encoding of unnatural amino acids (UAAs) allows the precise modulation of protein function within complex cellular processes. By incorporation of UAAs via suppression of the amber stop codon (TAG), target proteins can be site specifically modified[17–19]. Utilization of photo-caged UAAs, such as amino acids with attached *ortho*-nitrobenzyl or coumarin moieties, enable light-convertible states[20]. While the photocage initially renders the amino acid functionally inactive, illumination triggers uncaging and restores the native function. Thus, photo-caged UAAs are beneficial to gain control of protein activity within cellular pathways[21,22]. However, the major bottleneck for the utilization of UAAs is the engineering of an orthogonal aminoacyl-tRNA synthetase with a cognate tRNA that efficiently incorporates the UAA into the nascent polypeptide chain[23–25].

For the design of a photo-conditional TAP complex, we reasoned that the lysine residue in the Walker A motif of the canonical ATP-binding site would be best suited as it is critical for nucleotide binding. Thus, substitution of the lysine residue by a photo-caged UAA will initially block the coordination of the β- and γ-phosphate of ATP[26–29] and hence TAP-dependent antigen translocation. We utilized a lysine with a 6-nitropiperonyl cage, which can be cleaved off by illumination in the range of 365 nm[30,31]. Via light control, the native lysine function can be restored to activate TAP and the entire PLC (Fig. 1). In contrast to conventional knockout or knockdown approaches, which can be biased by complex compensation and feedback mechanisms[32], this photo-conditional system allows investigations on both, loss

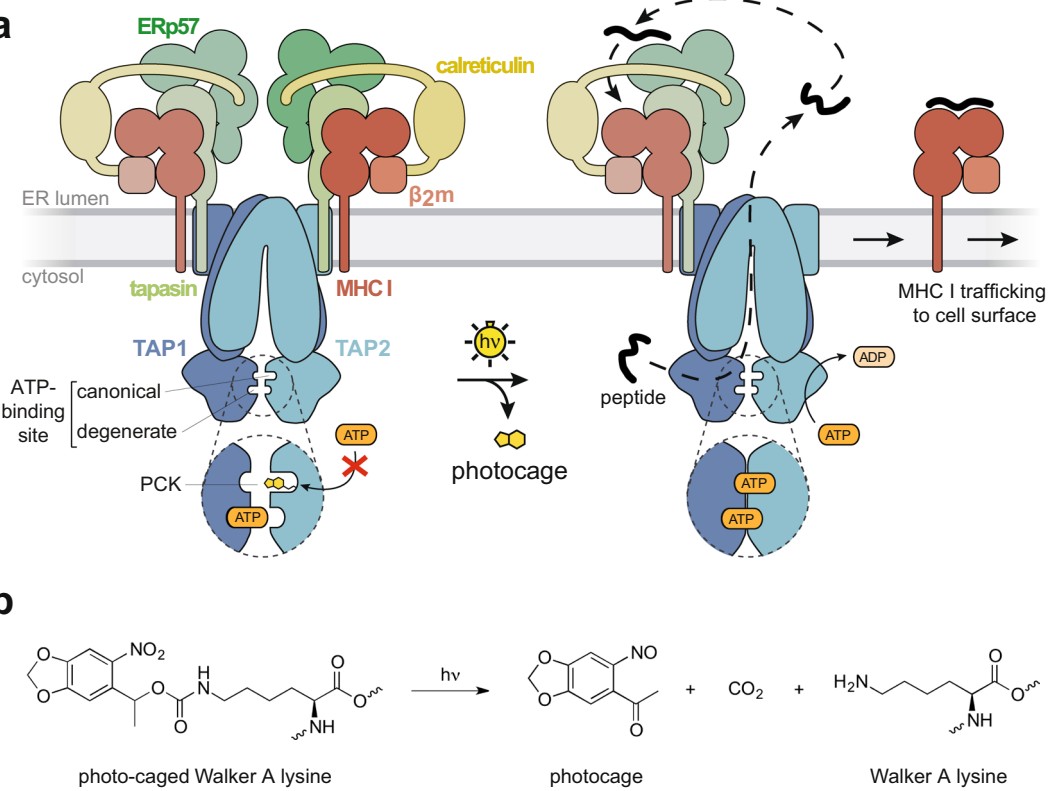

**Fig. 1 Photo-conditional peptide loading complex. a** The conserved lysine of the TAP2 Walker A motif, which is crucial for ATP binding and hydrolysis in the canonical ATP-binding site, is replaced by a photo-caged lysine (PCK) using amber suppression. TAP function is arrested due to the photocage that prevents ATP binding. After illumination and release of the photocage, TAP and PLC function is restored, which activates peptide translocation, MHC I loading and trafficking. **b** Illumination triggers the uncaging of PCK and allows restoration of the native lysine residue.

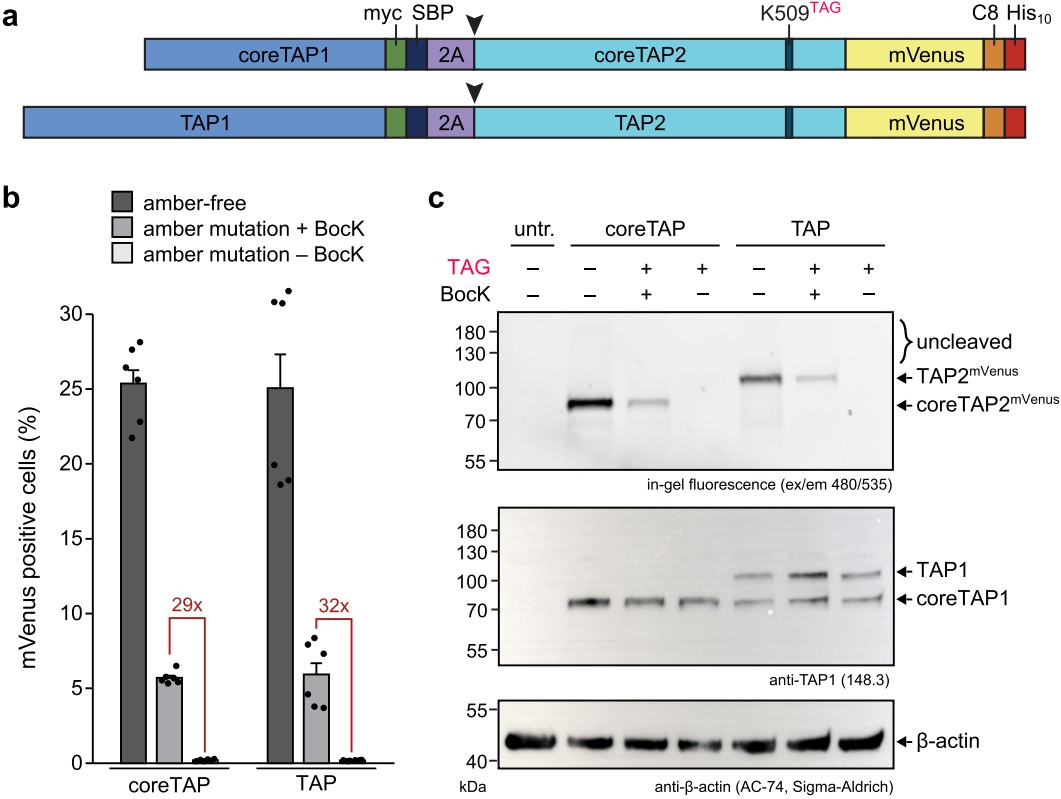

**Fig. 2 Design of amber-suppressed TAP constructs. a** Schematic representation of the TAP variants with amber mutation in TAP2 (K509, GPNGSG**K**ST). Separation into single TAP1 and TAP2 subunits encoded by an F2A site (purple) is highlighted by a black arrow. SBP streptavidin-binding peptide. **b** Co-expression of amber-free or amber-suppressed coreTAP or TAP with wtPylRS in the presence or absence of 250 μM BocK in TAP2-deficient STF1-169 cells analyzed by flow cytometry (mean ± SEM, $n = 6$, biologically independent samples). **c** F2A separation and amber-suppression efficiency for co-expression of coreTAP or TAP with wtPylRS in TAP2-deficient STF1-169 cells analyzed by SDS-PAGE, in-gel fluorescence (mVenus), and immunoblot (anti-TAP1 and β-actin). Constructs with amber mutation (TAG) were expressed with or without 250 μM BocK (representative data, $n = 2$, biologically independent samples). untr. untransfected cells.

and gain of protein function, in the same genetic and cellular context. By establishing an *in-cell* system of a photo-conditional TAP complex, we developed an approach, which provides new insights into antigen processing within the PLC and MHC I trafficking. This tight spatiotemporal control has not been achieved in the ABC superfamily or antigen presentation and is the first application in cellular immunology. Importantly, this system can be transferred to other NTPases with a phosphate-binding loop (P-loop) and the conserved lysine residue, e.g., G-proteins, the AAA + family, the ABC superfamily, the helicase superfamilies I, II, and III, thus expanding into a generic approach to explore various cellular pathways.

## Results

**Designing a photo-conditional gatekeeper of antigen presentation**. The Walker A motif, also termed P-loop, is essential for the function of basically all ATP/GTPases[33], including the ABC transporter TAP. Here, we amber suppressed the conserved lysine residue in the Walker A motif of TAP2 (K509, GPNGSG**K**ST), which is crucial for ATP binding and hydrolysis, thereby energizing the antigen translocation into the ER lumen. In addition to full-length TAP, we utilized also coreTAP, which lacks the extra N-terminal transmembrane TMD0 domains and codes solely for the core transporter unit. The coreTAP complex has been shown to be essential and sufficient for peptide binding and translocation[4]. Although PLC assembly requires the TMD0s, coreTAP expression can still restore 50% of the MHC I surface

expression in TAP-deficient cells[34]. Our expression constructs encoded both TAP subunits, which were separated by an F2A ribosome-skipping site, derived from the foot-and-mouth disease virus, and comprised either full-length TAP or coreTAP (TAP and coreTAP, respectively) (Fig. 2a). Utilizing the F2A site for TAP1/2 co-expression has three advantages: (i) minimizing the DNA load for facilitated single transfection conditions, (ii) a stoichiometric translation of both TAP subunits, and (iii) an increased stability of the subunits during their biosynthesis as TAP1 serves as chaperone for the unstable TAP2 subunit[35]. To simplify detection, coreTAP2 and TAP2 were C-terminally fused to mVenus.

We examined the expression of the coreTAP and TAP constructs in TAP-deficient fibroblast cells (STF1-169), which have been generated from a Bare-Lymphocyte syndrome patient[36] (Fig. 2b and Supplementary Fig. 1). CoreTAP or TAP with and without amber mutation was co-expressed with the wild-type pyrrolysyl-tRNA synthetase/tRNA$_{CUA}$ pair (from now on abbreviated as wtPylRS) derived from *Methanosarcina mazei*[37,38]. In the flow cytometric analysis both amber-suppressed TAP variants (from now on abbreviated as TAP$^{TAG}$ and coreTAP$^{TAG}$) showed an ~30-fold higher expression compared to the background without N$_\varepsilon$-Boc-L-lysine (BocK). Next, we analyzed expression and subunit separation of coreTAP and TAP on protein level (Fig. 2c). Apart from co-translationally processed coreTAP2$^{mVenus}$ and TAP2$^{mVenus}$, we did not observe any unprocessed expression products by in-gel fluorescence. Notably, amber-suppressed coreTAP2$^{mVenus}$ and TAP2$^{mVenus}$ were only detected in the

presence of BocK, while immunoblotting revealed similar coreTAP1 and TAP1 expression level for all conditions. Expression of full-length TAP1 resulted also in coreTAP1 as a natural cleavage product[4]. These results indicate both an efficient co-translational separation of the TAP subunits by the F2A site and successful amber suppression at position of the conserved lysine residue.

**A photo-caged lysine in the canonical ATP-binding site blocks TAP function**. For the specific incorporation of photo-caged lysine (PCK), we modified the *M. mazei* wtPylRS by mutating four residues (M276F, A302S, Y306C, L309M), which had been shown to enable PCK incorporation in the *M. barkeri* PylRS[30]. This engineered PylRS/tRNA$_{CUA}$ pair was named optPylRS. By co-expression of coreTAP$^{TAG}$ with optPylRS in the presence of different PCK concentrations, we examined the minimal concentration of PCK required to generate a photo-conditional TAP complex (Supplementary Fig. 2). We observed that for efficient amber suppression the PCK concentration can be decreased to

20 nM, which is substantially low compared to commonly applied UAA concentrations[30,31]. This careful optimization prevented not only wasting of the precious UAA but also minimized potential cytotoxic effects at high PCK concentrations. To verify the specific incorporation of PCK by optPylRS, we co-expressed coreTAP$^{TAG}$ with optPylRS or wtPylRS in TAP2-deficient cells and added either PCK, BocK, or no UAA. As control, amber-free coreTAP was co-expressed with optPylRS in the presence of PCK to assess the side effects of the UAA supplementation (Fig. 3a). Notably, only the co-expression of the coreTAP$^{TAG}$ construct with optPylRS in the presence of PCK or wtPylRS in the presence of BocK resulted in coreTAP$^{TAG}$ synthesis. No other combination of coreTAP$^{TAG}$ and PylRS with or without UAA led to the production of amber-suppressed coreTAP2. Hence, these PylRS/UAA combinations were exclusively utilized for PCK and BocK incorporation and abbreviated from now on as coreTAP$^{TAG}$/PCK and coreTAP$^{TAG}$/BocK. Compared to the amber-free construct, both amber suppression systems reached similar protein levels of 30 to 40% (Fig. 3a). Consequently, incorporation of the photo-caged

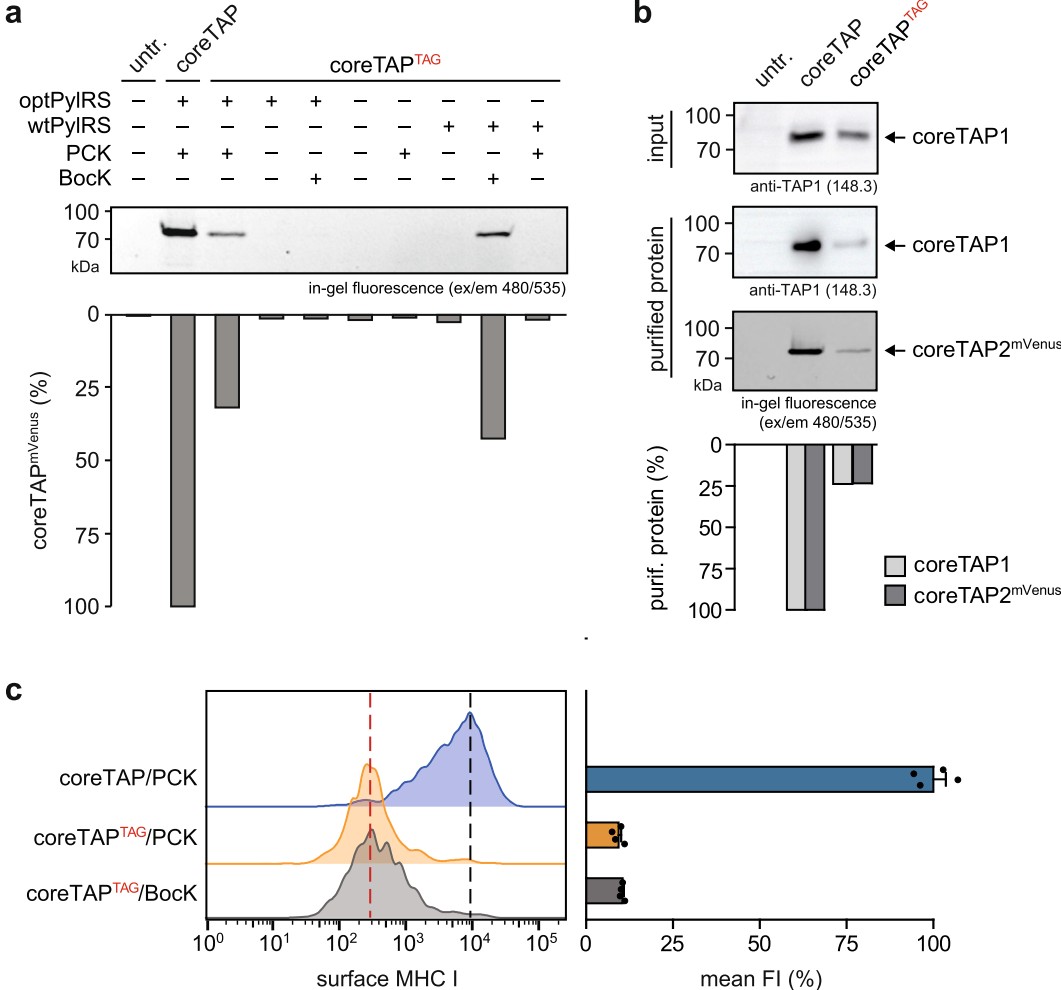

**Fig. 3 UAA incorporation in the canonical ATP-binding site of TAP blocks antigen presentation. a** TAP2-deficient STF1-169 cells co-expressing coreTAP$^{TAG}$ and wtPylRS or optPylRS in the presence of BocK or PCK were analyzed by SDS-PAGE and mVenus in-gel fluorescence. CoreTAP served as reference and was utilized to normalize the fluorescence intensities. untr. untransfected cells. **b** Purified coreTAP/PCK and coreTAP$^{TAG}$/PCK were analyzed by SDS-PAGE, in-gel fluorescence (mVenus), and immunoblot (anti-TAP1). To analyze the subunit stoichiometry of the purified TAP1/2 complex, the coreTAP1 signal of the input (cell lysate) was used for normalizing. **c** Surface MHC I expression in TAP2-deficient STF1-169 cells co-expressing coreTAP or coreTAP$^{TAG}$ and optPylRS in the presence of PCK or coreTAP$^{TAG}$ and wtPylRS in the presence of BocK. Flow cytometric analysis was performed by using an APC-Fire750-labeled HLA-A, B, C-specific antibody (W6/32). The black dotted line represents the mode of cells expressing coreTAP, while the red dotted line represents the mode FI of cells expressing the amber-suppressed variants. The mean FI of MHC I surface expression was normalized to coreTAP/PCK (± SEM, *n* = 4, biologically independent samples). FI fluorescence intensity.

lysine in the Walker A motif of coreTAP2 by optPylRS is efficient and specific for PCK.

To analyze the stoichiometric assembly of a photo-conditional TAP complex, coreTAP/PCK, and coreTAP[TAG]/PCK were expressed in HEK293-F cells to achieve higher transfection efficiencies and expression levels for isolation of the TAP1/2 complex. After orthogonal purification via His[10]-tag at TAP2 and streptavidin-binding peptide (SBP)-tag at TAP1, TAP complexes with a 1:1 stoichiometry were obtained, indicating a complete translation of the F2A constructs and assembly of the TAP subunits (Fig. 3b). Consistent to the expression levels in TAP-deficient cells, the yield of amber-suppressed coreTAP[TAG]/PCK is 25% of the amber-free construct. Next, we examined whether the site-specific UAA incorporation in the canonical ATP-binding site prevents TAP-dependent presentation of peptide-MHC I complexes on the cell surface (Fig. 3c). The MHC I surface expression was monitored by flow cytometry using a conformational specific anti-HLA-A, B, C antibody (W6/32). CoreTAP[TAG] was either expressed in the presence of PCK or BocK, while amber-free coreTAP/PCK served as control. Both amber-suppressed coreTAP variants displayed only 10% of MHC I surface expression compared to amber-free coreTAP (Fig. 3c), demonstrating that the incorporated PCK efficiently prevents MHC I trafficking via TAP inhibition.

**Optical control of antigen translocation**. After verifying TAP inhibition by PCK incorporation in the canonical ATP-binding site, we examined whether the function of photo-conditional TAP is restored by illumination. To this end, we monitored the TAP-dependent peptide transport into the ER using an ultra-sensitive single-cell assay[39] in TAP2-deficient cells expressing either coreTAP[TAG]/PCK or TAP[TAG]/PCK. Fluorescently labeled peptides were delivered into the cytosol after semi-permeabilization of the plasma membrane by the *Streptococcus pyogenes* toxin streptolysin O (SLO) (Fig. 4a). After optimization, we achieved 90% semi-permeabilization of the plasma membrane (Supplementary Fig. 3). By flow cytometry, we followed the antigen translocation of the peptide epitope RRYQNSTC[AlexaFluor647]L (NST[AF647]), which is retained in the ER lumen after N-core glycosylation (Fig. 4b, c and Supplementary Fig. 4). Cells expressing amber-free coreTAP or TAP were used as positive controls. Without illumination, these cells displayed an ATP-dependent peptide accumulation. Under the same conditions, cells expressing coreTAP[TAG] or TAP[TAG] in the presence of PCK or BocK did not show ATP-dependent peptide transport above background (mVenus mock control). Notably, after uncaging of TAP by illumination, we observed a light-induced peptide transport. TAP[TAG]/PCK and coreTAP[TAG]/PCK restored TAP-dependent transport activity to 75% and 32%, respectively, compared to the amber-free constructs (Fig. 4b, c). These results are in line with a lower expression of the amber-suppressed variants compared to the amber-free constructs (Figs. 2, 3, Supplementary Figs. 1, 3) and demonstrate that the function of the antigen translocation complex TAP can be controlled in a precise manner via a single photo-caged amino acid.

**Light-triggered MHC I trafficking and surface presentation**. Encouraged by the finding that TAP-dependent antigen translocation can be controlled by light, we investigated whether this approach can be utilized to understand how PLC activation is coupled to MHC I trafficking and antigen presentation. To this end, we monitored peptide–MHC I complexes emerging at the cell surface after illumination. The MHC I surface expression was analyzed by flow cytometry using an HLA-A, B, C-specific antibody as described above. As first reference, we used an 'acid wash'[40] and determined an appropriate timescale of 4 h for the

analysis of newly surfaced peptide–MHC I complexes (Supplementary Fig. 5). Light activation of both, coreTAP[TAG]/PCK and TAP[TAG]/PCK, induced a significant increase in MHC I surface expression compared to non-illuminated cells, while MHC I presentation was not altered immediately after illumination (Fig. 5a, b). Light activation of full-length TAP[TAG] resulted in a slightly higher MHC I surface level compared to coreTAP[TAG] (Fig. 5a, b), reflecting that photo-conditional full-length TAP promotes the optimal assembly of the PLC[34]. Interestingly, even before illumination, TAP[TAG] expression led to an elevated MHC I surface presentation compared to coreTAP[TAG], indicating that photo-conditional TAP[TAG] can assemble a symmetric PLC and thereby promote the MHC I loading of TAP-independent epitopes. By following the light-induced MHC I surface presentation of coreTAP[TAG]/PCK- and TAP[TAG]/PCK-expressing cells over time, we observed a gradual increase with a lag phase of 1 h (Fig. 5c, d). Based on the optical control of TAP-dependent antigen processing, we can temporally separate PLC assembly from peptide translocation and MHC I trafficking. These results do not only demonstrate the precise control of MHC I antigen presentation by photo-conditional TAP but also the potential impact of pre-assembled multi-subunit complexes and cellular pathways synchronized via light.

**Discussion**

Here, we describe an approach for the spatiotemporal analysis of TAP-dependent peptide translocation, loading, and MHC I trafficking via light control. We genetically encoded the photo-caged lysine PCK in the Walker A motif of TAP2, which is essential for the peptide-transport function of TAP within the PLC. The site-specific incorporation of PCK required an orthogonal system, co-expressing the PCK optimized aminoacyl-tRNA synthetase optPylRS, its cognate tRNA, and amber-suppressed TAP. Photo-conditional TAP was exclusively expressed in the presence of optPylRS and PCK, demonstrating specific expression of the amber-suppressed TAP2 subunits (Fig. 3a). In contrast to previously reported photo-caged UAA approaches[30,31,37], PCK was sufficiently incorporated if supplied at nanomolar concentrations (Supplementary Fig. 2). Thus, we could minimize possible cyto-toxic effects and express photo-conditional TAP under near-physiological conditions. To gain new insights into TAP function within the PLC, we selected constructs coding for full-length TAP or coreTAP separated by an F2A ribosome-skipping site to produce both TAP subunits in a native 1:1 stoichiometry (Fig. 3b). By utilizing a single-cell-based peptide translocation assay[39] combined with monitoring MHC I surface expression, we demonstrated that PCK incorporation in the canonical ATP-binding site blocks TAP function (Figs. 3c and 4b, c). Illumination triggered antigen translocation into the ER lumen (Fig. 4b, c), followed by MHC I trafficking and antigen surface presentation (Fig. 5). Light activation of both, photo-conditional full-length TAP and coreTAP, induced a gradual accumulation of newly surfaced MHC I molecules. These results are consistent with previous findings demonstrating that an asymmetric PLC can be assembled on endogenous full-length TAP1 and coreTAP2[34]. However, illumination of full-length TAP led to a slightly higher surface MHC I presentation compared to coreTAP. Interestingly, this trend for an elevated MHC I level of full-length TAP expressing cells had already been seen before light activation. This result illustrates the important function of TAP as a PLC scaffold and indicates beneficial effects of PLC assembly even without TAP-dependent antigen translocation.

Since MHC I trafficking and turnover is still poorly characterized, we utilized an 'acid wash' as first reference of the MHC I restoration at the cell surface[40]. However, it is important

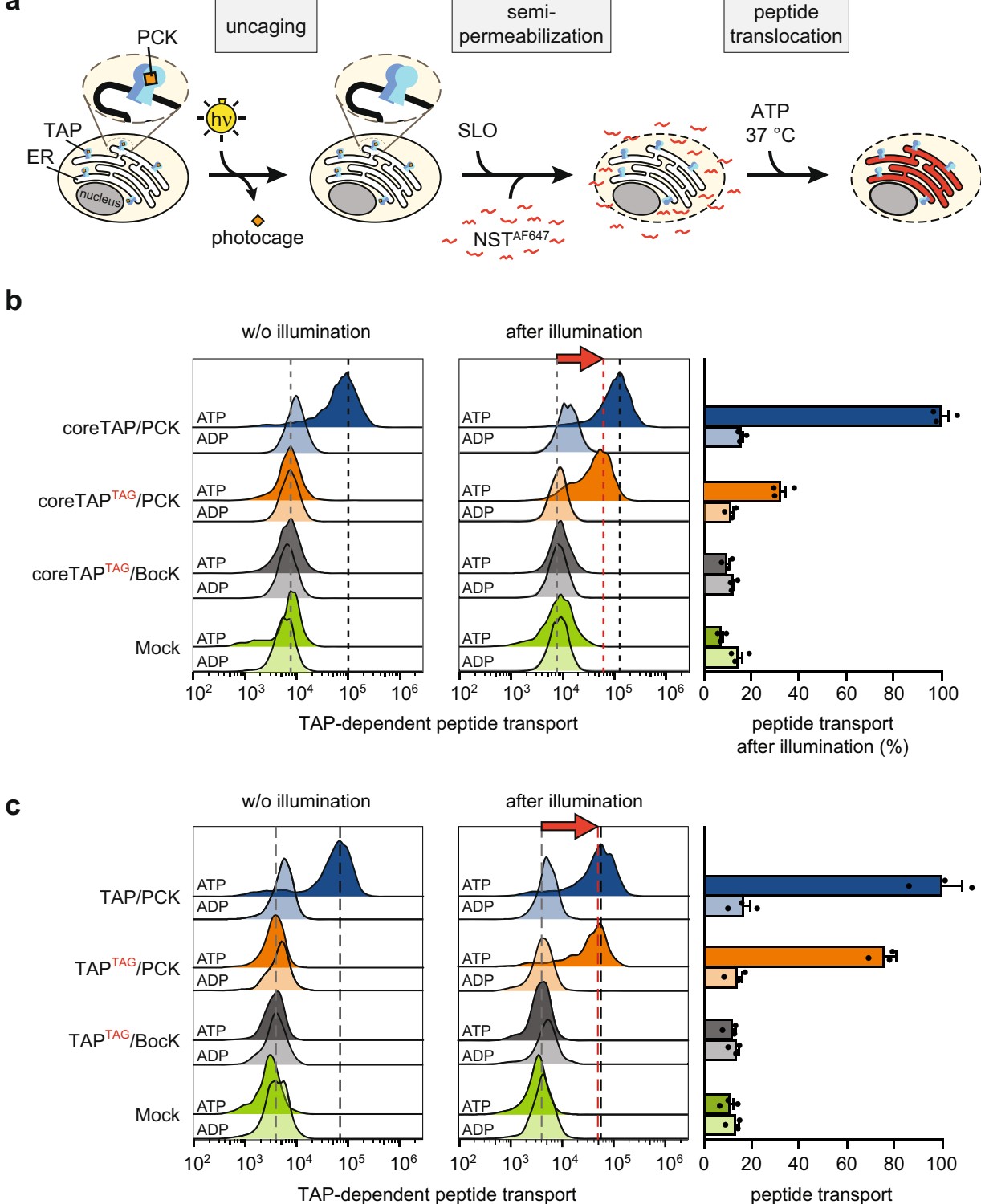

**Fig. 4 Antigen translocation triggered by light. a** Schematic representation of TAP-dependent peptide translocation. TAP2-deficient STF1-169 cells expressing photo-conditional TAP were illuminated and semi-permeabilized with streptolysin O (SLO). Peptide translocation was performed at 37 °C in the presence of fluorescently labeled NST$^{AF647}$ peptide and ADP or ATP. **b** Peptide translocation of TAP2-deficient STF1-169 cells expressing coreTAP or **c** TAP variants without and with illumination was monitored by flow cytometry. mVenus expression served as mock control. The dotted lines represent the mode FI of the transported peptide of the corresponding amber-free counterpart (black) and the photo-conditional variants before (gray) and after illumination (red). Peptide transport after illumination was determined by normalizing the mean FI (± SEM, $n = 3$, biologically independent samples) of transported peptide to the corresponding amber-free variant in presence of ATP. FI fluorescence intensity.

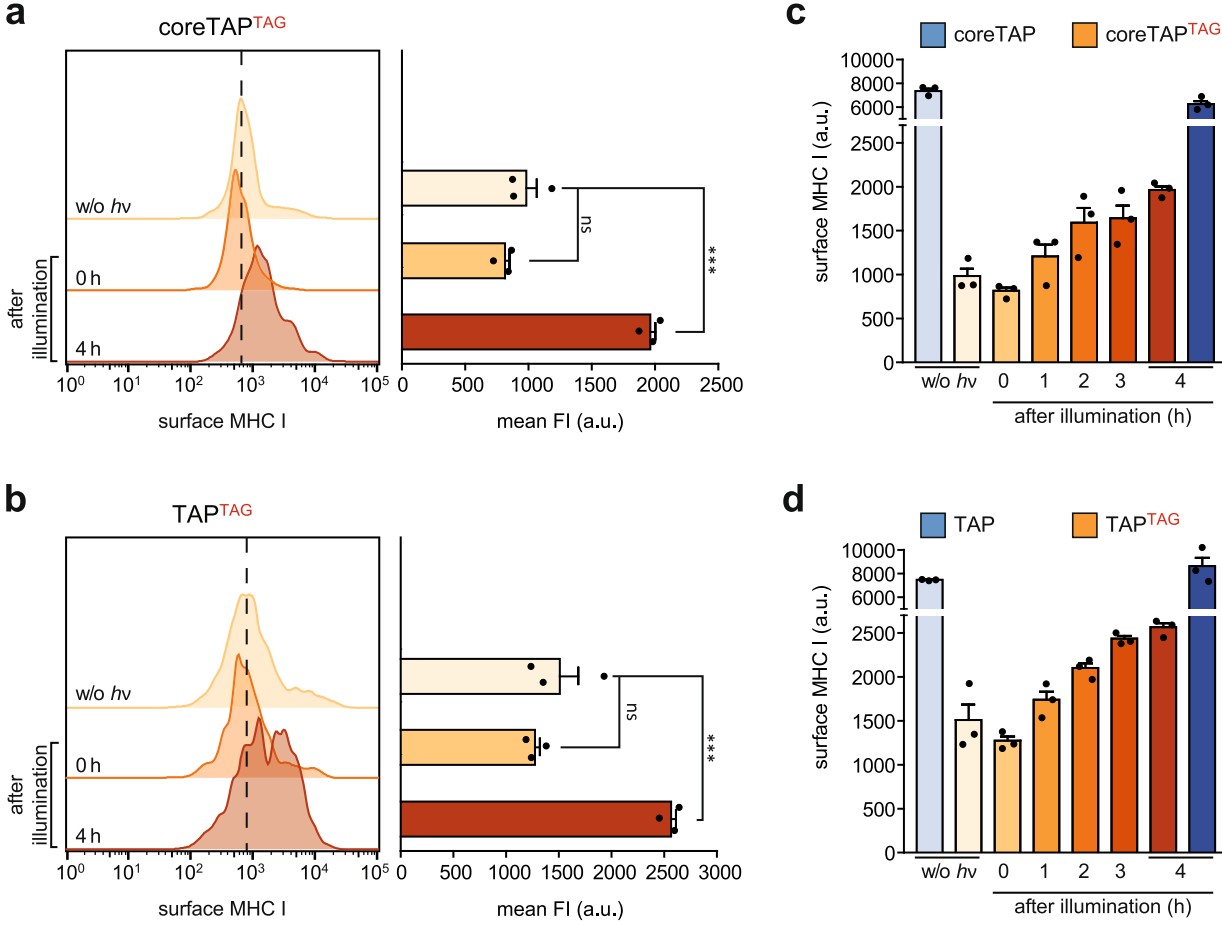

**Fig. 5 Light activation of TAP synchronizes MHC I trafficking and surface presentation. a** coreTAP$^{TAG}$/PCK and **b** TAP$^{TAG}$/PCK was expressed in TAP2-deficient STF1-169 cells. MHC I surface expression of the mVenus-positive cells was monitored before (w/o $h\nu$) and after illumination by flow cytometry using an APC-Fire750-labeled HLA-A, B, C-specific antibody (W6/32). The black dotted line represents the mode FI of non-illuminated cells. Mean FI was calculated (±SEM, $n = 3$, biologically independent samples) and a One-way ANOVA with Turkey's multiple comparison test was performed. ns non-significant; ***, $p < 0.0001$. **c** coreTAP$^{TAG}$/PCK- and **d** TAP$^{TAG}$/PCK-dependent MHC I surface expression (± SEM, $n = 3$, biologically independent samples) up to 4 h after illumination. FI fluorescence intensity.

to note that the intracellular pools of MHC I are not affected, and peptide–MHC I complexes can still continuously traffic to the cell surface. In contrast, our approach to control TAP and PLC function by light offers the advantage to trigger peptide translocation and MHC I loading in a spatiotemporal manner and provides a well-synchronized starting point of MHC I trafficking. Combined with activation via confocal laser illumination, photo-conditional TAP has the potential to open avenues for studies on various intracellular MHC I pools and antigen presentation pathways in different cells. However, achieving levels of the amber-free variants in functional assays might not be possible, as we showed by the partial restoration of the peptide translocation activity (Fig. 4b, c) and MHC I surface presentation (Fig. 5c, d). Besides the reduced expression levels compared to their amber-free counterparts, a possible explanation for this result is that active TAP, which is exclusively generated by illumination, is subject to protein turnover. After light activation, de novo TAP2 expression is prematurely terminated by the amber stop codon. Thus, degradation of light-activated TAP, depending on the half-life of the variants, might be relevant for experiments that have to be monitored over long periods of time.

Conventional methods to investigate complex cellular processes such as antigen processing and presentation utilize inhibitors or conditional knockdown/knockout of key players. Nevertheless, both strategies are incapable to restore function of the target

protein and, therefore, do not represent an ideal approach. As inhibitors allow the arrest of protein activity while maintaining the native cellular environment, we previously reported a method to arrest and re-activate TAP via a photo-conditional variant of the viral inhibitor ICP47[41]. However, this approach has one major drawback: the photo-conditional immune suppressor is a synthetic cell-impermeable compound, which requires delivery into the cytosol. In contrast, the genetic encoding of an initially arrested TAP enables studies on the loss of function within an assembled PLC as well as antigen translocation and MHC I presentation after light activation.

In conclusion, we established the optical control of TAP-dependent antigen processing while maintaining the native PLC assembly. Thus, photo-conditional PLC enables precise control of antigen translocation, peptide loading onto MHC I molecules, and surface presentation of peptide–MHC I complexes. In contrast to conventional knockout or knockdown approaches, TAP-dependent and TAP-independent antigen presentation can be dissected since the PLC, including MHC I assembly and peptide editing, remains intact[42,43]. These advantages open avenues for new in vitro and in vivo studies in the field of antigen presentation, such as time-resolved investigations on PLC dis-assembly and MHC I trafficking in different presentation pathways. Moreover, this approach can be versatilely transferred to various other ATP/GTPases[21,22,31], especially in the context of

protein complex formation within cellular pathways. Hence, this system has the potential to answer key questions of intra- and intercellular processes that are essential for cell homeostasis and spatiotemporal compartmentalization of antigens.

## Methods

**Synthesis of nitropiperonyl caged lysine (PCK).** All reagents and solvents were of the best grade available supplied by Fluka, Sigma Aldrich, Merck, or Carl Roth and used without further purification. Boc-Lys-OH was obtained from Iris Biotech. Unless otherwise stated, all reactions were performed under argon atmosphere using dry solvents (Sigma Aldrich). Analytical thin-layer chromatography was performed on silica gel 60 plates with fluorescence indicator (254 nm, Merck KGaA). Column chromatography was performed on silica gel 60 (40–63 μm, Macherey-Nagel GmbH & Co. KG) or silica gel C18 end-capped (0.035–0.07 mm, 400–220 mesh, Carl Roth) using solvents of technical grade. Synthesis of PCK (Supplementary Fig. 6) was modified according to Gautier and colleagues[30]. ESI-MS spectra were recorded on VG Fisons with quadrupole analyzer. [1]H-NMR spectra were recorded on a BRUKER AV250 or AV400. NMR-Signals were calibrated on solvent signals: [1]H-NMR: CDCl$_3$:7.26 ppm; D$_2$O:4.79 ppm. [1]H-NMR data are presented as follows: chemical shift in ppm (multiplicity, coupling constant, integration). The following abbreviations are used in reporting NMR-Data: s singlet, d doublet, t triplet, q quartet, dd doublet of doublets, m multiplet.

*2-(tert-Butoxycarbonylamino)-6-([1-(6-nitrobenzo[d][1,3]dioxol-5-yl)ethoxy]carbonylamino)hexanoic acid (3).* 1-(6-Nitrobenzo[d][1,3]dioxol-5-yl)ethanol **1** (500 mg, 2.37 mmol) was dissolved in THF (4 ml), DIPEA (2.5 ml, 14.22 mmol) was added and cooled to 0 °C. After adding triphosgene ((bis(trichloromethyl) carbonate) (1.06 g, 3.56 mmol), the reaction was stirred for 30 min at 0 °C and then allowed to stir 12 h at RT. The reaction was filtered and the solvent was evaporated without heating to give nitropiperonyl chloroformate **2** in quantitative conversion (648 mg, 2.37 mmol). A solution of N$_ε$-Boc-lysine (1.17 g, 4.74 mmol) in water (4 ml) containing DIPEA (2.5 ml, 14.22 mmol) was added dropwise to a solution of chloroformate **2** in 1,4-dioxane (4 ml) at 0 °C. The reaction was stirred for 30 min at 0 °C and then continued 12 h at RT. The solution was acidified with 1 M HCl to pH 1 and extracted with Et$_2$O. Combined organic phases were dried over Na$_2$SO$_4$, then filtered, and the volatiles were evaporated. The crude product was purified by column chromatography using RP-C$_{18}$ silica H$_2$O/MeCN (1:1) yielding **3** as a yellow solid (51% yield, 585 mg, 1.21 mmol). [1]H NMR (300 MHz, CDCl$_3$) δ = 1.32–1.68 (m, 18 H), 3.13 (br s, 2 H), 4.28 (br s, 1 H), 4.90 (br s, 1 H) 5.13 (br s, 1 H), 6.10 (s, 2 H), 6.24–6.28 (m, 1 H), 7.00 (s, 1 H), 7.47 (s, 1 H). HR ESI-MS: m/z calculated for C$_{21}$H$_{28}$N$_3$O$_{10}$ [M$^-$]: 482.18; found: 482.27.

*2-Amino-6-([1-(6-nitrobenzo[d][1,3]dioxol-5-yl)ethoxy]carbonylamino)hexanoic acid TFA salt (4).* Compound **3** (500 mg, 1.03 mmol) was dissolved in DCM:TFA (1:1 mixture, 14 ml total), and the reaction was allowed to stir for 2 h. The volatiles were evaporated and the residue was purified by column chromatography using RP-C$_{18}$-silica H$_2$O/MeCN (1:1) yielding 32% **4** as a white solid (126 mg, 0.33 mmol). [1]H NMR (250 MHz, D$_2$O) δ = 1.04–1.37 (m, 9 H), 1.57–1.75 (m, 2 H), 1.81 (d, 1 H), 2.80 (m, 2 H), 3.71 (m, 1 H), 5.83 (d, 1 H), 5.89 (s, 2 H), 6.87 (s, 1 H), 7.28 (s, 1 H). HR ESI-MS: m/z calculated for C$_{16}$H$_{22}$N$_3$O$_8$ [M + H]$^+$: 384.14; found: 384.06.

**Expression constructs.** All TAP constructs are based on de novo synthesized sequences (NCBI Reference Sequence: NP_000584.3, residues 1–748, and 164–748 for TAP1 and coreTAP1, respectively; NP_000535.3 incl. A665T substitution found in TAP2 short allele CAA80523.1, residues 1–703 and 123–703 for TAP2 and coreTAP2, respectively), where all cysteines except C213 in TAP2 are replaced. This single intrinsic cysteine is required to maintain a peptide-receptive conformation[44,45]. F2A ribosome skipping site was adapted from Minskaia and colleagues[46] and de novo synthetized, containing 58 residues VTELLYRMKRAE-TYCPRPLLAIHPSEARHKQKIVAPVKQLLNFDLLKLAGDVESNPGP. All cloning steps and inserted mutations were based on PCR amplification by *Pfu* DNA polymerase with primers generating LguI restriction sites (TAP2 K509TAG fw 5′-TATATAGCTCTTCTTAGTCCACCGTCGCTGCTC-3′, TAP2 K509TAG rev 5′-TATATAGCTCTTCTCTAGCCAGAACCGTTAGGGC-3′, PylRS M276F fw 5′-TATATAGCTCTTCTTTCGGCATCGACAACGACACC-3′, PylRS M276F rev 5′-TATATAGCTCTTCTGAACCGCTGATGTACTCCAG-3′, PylRS Y306C L309M fw 5′-TATATAGCTCTTCTAACTACATGCGGAAACTGGATCGCG C-3′, PylRS Y306C L309M rev 5′-TATATAGCTCTTCTGTTGCACAGGTT GGGGGC-3′, PylRS A302S fw 5′-TATATAGCTCTTCTAGCCCCAACCTG TGCAACTAC-3′, PylRS A302S rev 5′-TATATAGCTCTTCTGCTCAGCATGG GCCGC-3′). DNA fragments were ligated in a one-pot reaction with fast digest LguI (Thermo Fisher Scientific), 1x FD buffer, 0.5 mM ATP and T4 DNA ligase (Thermo Fisher Scientific). Constructs coding for coreTAP1-myc-SPB-F2A-coreTAP2-mVenus-C8-His$_{10}$ and TAP1-myc-SPB-F2A-TAP2-mVenus-C8-His$_{10}$ were generated in a pcDNA backbone. The plasmid encoding for wild-type PylRS derived from *Methanosarcina mazei*[37,38] and four copies of tRNA$_{CUA}$ were kindly provided by Dr. Katrin Lang, TU Munich, Germany, and allowed the incorporation of N$_ε$-Boc-L-lysine. To simplify mutagenesis, the PylRS was transferred to an empty pET SUMO backbone. After substitution of four residues (M276F, A302S, Y306C, L309M) that are essential for the specific binding of PCK ((N6-[[1-(6-nitro-1,3-benzodioxol-5-yl)ethoxy]carbonyl]-L-lysine), this optimized PylRS (opt-PylRS) was transferred back to the initial backbone. All constructs were verified by DNA sequencing.

**Cell lines and culture.** The STF1-169 cell line (HLA-A*03:01, HLA-B*15:16, HLA-C*14:02) was kindly provided by Dr. Henri de la Salle, University of Strasbourg, France[36], and cultured in DMEM (Gibco) supplemented with 10% fetal calf serum (FCS, Gibco) at 37 °C and 5% CO$_2$. Adherent FreeStyle™ 293-F cells (HEK293-F cells) were cultured in DMEM (Gibco) supplemented with 10% FCS at 37 °C and 5% CO$_2$ and in FreeStyle™ 293 Expression Medium (Gibco) at 37 °C, 8.1% CO$_2$, and 125 rpm for suspension culture.

**Transfection of TAP2-deficient STF1-169 cells.** $1.5 \times 10^4$ STF1-169 cells per well were seeded in 12-well plates. After 16 h, cells were transfected using Lipofectamine LTX (Thermo Fisher Scientific). 1.5 μg DNA and 1.5 μl Plus reagent were mixed in 100 μl Opti-MEM (Gibco). 2.5 μl Lipofectamine LTX were mixed with 100 μl Opti-MEM. For transfection in a 6-well format, double the number of cells and transfection mixture were applied. The two solutions were mixed and incubated 30 min at RT. After medium exchange, the transfection solution was added dropwise. 5.5 h after transfection, the medium was exchanged for DMEM supplemented with 10% FCS. Cells used for incorporation of UAAs were further supplemented with 0.25 mM BocK or 20 nM PCK along with the medium change after transfection.

**Transfection of HEK293-F cells.** $1.5 \times 10^8$ HEK293-F cells were seeded in 300 ml FreeStyle™ 293 Expression Medium. After 24 h, cells were transfected using PEI. 300 μg DNA and 1.2 mM PEI (1 mg/ml) were dissolved separately in 10 ml Opti-MEM I (1X) + GlutaMAX™ reduced serum medium (Gibco). The solutions were mixed and incubated for 30 min at RT and then added to the flasks. Cells used for incorporation of UAAs were further supplemented with 20 nM PCK. The cells were harvested 30 h after transfection.

**In-gel fluorescence and immunoblotting.** The transfected cells of one 6-well plate were harvested and lysed in 30 μl Pierce RIPA buffer (Thermo Fisher Scientific) supplemented with 1% benzonase (Novagen, EMD Chemicals) and subsequently incubated for 2 h at RT. Sodium dodecyl sulfate (SDS)-buffer was added to a final concentration of 62.5 mM Tris/HCl, pH 6.8, 100 mM β-mercaptoethanol, 2% SDS, 0.02% bromophenol blue, and 10% glycerol. 30 μl of sample were loaded on a 10% SDS-PA-gel (Laemmli) and run for 1 h at 125 V. In-gel fluorescence was detected using a Fusion FX (Vilber) at λ$_{480/535}$ for mVenus detection. Protein samples used solely for immunoblotting were heated for 10 min at 62 °C. Blotted PVDF membranes were blocked with 5% milk in TBS-T and incubated with anti-TAP1 148.3 (hybridoma supernatant generated in-house, 1:10)[47] and anti-β-actin (Sigma, clone AC-74, 1:5000). Anti-mouse IgG (Fc specific)-Peroxidase conjugate (Sigma, 1:20,000) was used as secondary antibody. For detection, membranes were incubated with Clarity Western ECL reagent (BioRad) or LumiGLO® Peroxidase Chemiluminescent Substrate Kit (seracare), and chemiluminescence was measured with a Fusion FX (Vilber). Quantification of signal intensities was done with ImageJ (1.52a).

**Purification of TAP.** For the orthogonal purification of coreTAP complexes, all steps were carried out on ice, and all buffers were adjusted to pH 7.4. The transiently transfected and harvested HEK293-F cells were solubilized for 1.5 h in buffer 1 (20 mM HEPES/NaOH, 200 mM NaCl, 50 mM KCl, and 15% (v/v) glycerol) supplemented with 10 mM imidazole, 1× PI-mix HP (Serva), and 2% (w/v) glyco-diosgenin (GDN, Anatrace). The samples were centrifuged for 30 min at 120,000 × *g* and 4 °C. The proteins were bound to 200 μl Ni-NTA Sepharose 6 Fast Flow (GE Healthcare) for 2 h. The beads were washed twice for 15 min with buffer 1 supplemented with 10 mM imidazole and 0.05% GDN. To elute the proteins, the beads were incubated for 30 min in buffer 1 supplemented with 200 mM imidazole and 0.05% GDN. The eluate was incubated 3 h with 200 μl high-capacity streptavidin agarose resin (Thermo Fisher Scientific). The beads were washed twice for 15 min in buffer 1 supplemented with 0.05% GDN. Subsequently, bound proteins were eluted for 45 min in buffer 1 supplemented with 2.5 mM biotin and 0.05% GDN. The eluate was frozen in liquid nitrogen and stored at −80 °C.

**Light activation.** Growth medium was replaced by preheated PBS (37 °C) supplemented with 10% FCS. Then, the culture plates (6- or 12-well) were placed on a transilluminator (312 nm, 12 mW/cm$^2$) and illuminated three times for 2 min with 2 min breaks in between to prevent the medium from overheating. After illumination, PBS was replaced by DMEM supplemented with 10% FCS. The cells were cultured in the dark at 37 °C and 5% CO$_2$ until further use.

**MHC I surface presentation.** MHC I surface expression was analyzed by flow cytometry. TAP2-deficient STF1-169 cells were harvested 18 to 24 h after complementation with different TAP constructs. All further steps were carried out on ice. The cells were washed in FACS-buffer (2% FCS in PBS, ice-cold) and

centrifuged for 5 min at 300 × g and 4 °C. After discarding the supernatant, cells were blocked for 15 min by 5% BSA in FACS-buffer. Subsequently, the cells were washed in FACS-buffer and stained for 20 min with 0.75 µl APC-Fire750 anti-human HLA-A, B, C (clone W6/32; BioLegend) in 50 µl FACS-buffer. Finally, cells were washed and resuspended in FACS-buffer for analysis by flow cytometry. Data was recorded by a FACSCelesta (BD) and processed using FlowJo V10 software. One-way ANOVA with Turkey's multiple comparison test was performed using GraphPad Prism 5. To determine the incubation time for analysis of MHC I surface expression, an acid treatment was performed. Twenty-four hours after complementation with coreTAP, surface pMHC I complexes were denatured for 2.5 min with 50 mM sodium citrate buffer, pH 3.0, and neutralized using three volumes of 150 mM $Na_2HPO_4$, pH 10.5[40]. After washing, cells were stained in 45 µl 5% BSA in FACS-buffer mixed with 5 µl biotinylated anti-human HLA-A, B, C (clone W6/32; BioLegend) for 20 min on ice to mask any remaining peptide–MHC I complexes and to block the cell surface. Cells were washed, resuspended in medium, seeded in 12-well plates, and incubated at 37 °C and 5% $CO_2$. After washing, cells were stained for 20 min on ice with 1 µl APC-Fire750 anti-human HLA-A, B, C in 50 µl 5% BSA in FACS-buffer. Finally, cells were washed and resuspended in FACS-buffer for analysis by flow cytometry. Data was recorded by a FACSCelesta (BD) and processed using FlowJo V10 software.

**Peptide transport assay**. STF1-169 cells were utilized for the flow cytometry TAP peptide transport assay. To determine the optimal SLO concentration for semi-permeabilization, $5 \times 10^4$ cells were seeded in 12 wells 24 h in advance. Cells were washed with PBS and then resuspended in 50 µl AP-buffer (PBS supplemented with 10 mM $MgCl_2$) with different amounts of SLO (1, 2, 4, 6, 8, and 10 ng/µl; 1 ng ≙ 0.55 U). After 15 min incubation on ice, 100 µl AP-buffer was added. The cells were immediately analyzed by flow cytometry. The transport assay was performed with transiently transfected cells (24 h after transfection). The cells were semi-permeabilized for 15 min on ice with 10 ng/µl SLO in 50 µl AP-buffer. Subsequently, the peptide transport was carried out for 15 min at 37 °C in 50 µl AP-buffer supplemented with 30 nM NST$^{AF647}$ peptide (RRYQNSTC$^{AF647}$L; AF647, Alexa Fluor™ 647) and 10 mM ATP or ADP. RRYQNSTCL was synthesized by Fmoc-SPPS (LibertyBlue, CEM). After peptide deprotection and cleavage, it was labeled with Alexa Fluor 647 C2 maleimide (Thermo Fisher Scientific). RRYQNSTC$^{AF647}$L was purified by C18 RP-HPLC. Purity and identity were verified by LC-MS (BioAccord, Waters). $M_{calc}$ 2120.7896 Da, $M_{obs}$ 2120.7854 Da. Peptide transport was stopped by adding 150 µl PBS supplemented with 20 mM EDTA. Data was recorded by a FACSCelesta (BD) and processed using FlowJo V10 software. Gating strategy for semi-permeabilized cells was adopted as previously described[48]. Significance of peptide accumulation was calculated by One-way ANOVA with Turkey's multiple comparison test using GraphPad Prism 5.

**Statistics and reproducibility**. The number of biologically independent samples and statistical tests are provided in the corresponding figure legends. Statistical analysis was performed by using GraphPad Prism 5.

**Reporting summary**. Further information on research design is available in the Nature Research Reporting Summary linked to this article.

## Data availability

All data are accessible from the corresponding author upon reasonable request. Raw data underlying any graph are provided as a source data file (Supplementary Data 1). Source data underlying the gels/blots and plasmid maps are included in Supplementary Data 2.

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

## Acknowledgements

We thank Andrea Pott, Inga Nold, Dr. Lukas Sušac, Dr. Christoph Thomas, Dr. Simon Trowitzsch, and all members of the Institute of Biochemistry for discussion and comments. We are grateful to Dr. Henri de la Salle (University of Strasbourg, France) for providing the STF1-169 cell line and Dr. Kathrin Lang (Technical University of Munich, Germany) for providing the plasmid encoding for *M. mazei* wild-type PylRS and four copies of tRNA$_{CUA}$. We thank Dr. Halvard Bönig (Goethe University Frankfurt) for HLA genotyping of the STF1-169 cell line. The work was supported by the German Research Foundation (GRK 1986 to R.W. and R.T.; TA 157/12-1 and CRC 807—Membrane Transport and Communication to R.T.), as well as by the European Research Council (ERC Advanced Grant 789121 to R.T.).

## Author contributions

J.B., V.H., and E.G.C. performed the experiments. H.K. and R.W. were involved in the synthesis and analysis of the caged amino acid. J.B., V.H., E.G.C., and R.T. designed the experiments and analyzed the data. J.B., V.H., and R.T. wrote the manuscript. R.T. conceived the study and supervised the project.

## Funding

## Competing interests

The authors declare no competing interests.
