## [Peer Review File · Communications Biology]

Reviewers' comments:

Reviewer #1 (Remarks to the Author):

The authors have established a light-controlled system to study the part of the antigen processing chain starting with transport of peptides by TAP. The TAP peptide transport is essential for efficient loading of peptides onto MHC class I molecules and after quality control of the resulting peptide-MHC-class I complexes presentation of the peptides to the immune defence cells on the cell surface. This paper shows that the method using an unnatural amino acid: photo-caged nitrioperonyl caged lysine (PKC) in the catalytic ATP-binding site of TAP allows control of the peptide transport step taking peptides from the cytosol to the ER lumen and the waiting MHC class I molecules. The method presented in the paper provides a very useful tool that if successfully applied can be used to widely study PLC assembly separated from peptide translocation and MHC class I transport.

This paper is a nice continuation from previous work, including the work done where TAP was arrested and re-activated through a photo-conditional variant of the viral inhibitor ICP47. The method presented will be of importance for immunologists and others wanting to advance the understanding of antigen presentation on a mechanistic level as well as in both health and disease. Many viruses and tumors target components of the antigen processing machinery, including the TAP proteins per se, and the consequences are of relevance not only for understanding the mechanisms and principles but also for finding and developing approaches for immunotherapy and selection of antigenic peptide candidates.

The data are collected and analysed with a well-planned and careful approach. However, since the type of HLA-I alleles vary in dependence on both the quality control system and the TAP transported peptides it would be appropriate and relevant if the HLA-I alleles of the model cell lines was written out in the material and methods. If not known, the cell lines should be typed and the information provided

The use of statistics seem appropriate but for the reader it could be more clearly explained e.g., what does n refer to and how many times were the experiments repeated with similar outcome. Fig. 3. Was the FACS done in quadruplicates or is the panel to the right four sets pooled from different experiments? Fig. 5. is n, the number of experiments performed at different occasions? Or is n the number of replicates in one experiment?

The conclusions are well-balanced and justified by the data presented in the paper.

Suggested improvements: experiments, data for possible revision

The paper is solid and interesting. No further experiments or additional data analysis needed, if the manuscript is complemented with information about the HLA-I alleles in the cell lines.

The references are appropriate and credit previous work in the field.

The abstract could perhaps be improved by emphasis on the functionality of the assay and thereby also justify the choice and development of method.

In the discussion it would be interesting if the authors related to if it plausible that there are mechanisms for compensation, i.e., increased egress rate or similar that operate when MHC class I molecules are washed away from the cell surface? Additional studies on allelic differences would be

of importance and not least, on combination of different HLA-I haplotypes.

/Kajsa M. Paulsson, Lund University

Reviewer #2 (Remarks to the Author):

The manuscript "Light control of the peptide-loading complex synchronizes antigen translocation and MHC I trafficking" by Brunnberg et al. reports a novel method for triggering the activity of Transporter associated with Antigen Processing (TAP) with light. TAP transports cytosolic peptides into the endo-plasmic reticulum so that Major Histocompatibility Complex I (MHC I) can display them on the cell sur-face, activating the immune system. To attain control over TAP, the authors used genetic code expan-sion to mutate residue K511 in TAP to an unnatural amino acid, the photocaged lysine PCK. This resi-due comprises lysine in which the side chain amine is masked with a photocleavable 6-nitropiperonyl protecting group via a carbamate linkage. K511 is essential for TAP to bind to and hydrolyze ATP. Thus, the TAP-K511PCK mutant is inactive until it is converted back the wild-type protein using UV light.

The authors devised a system using a ribosome-skipping site to produce stoichiometric quantities of the TAP1 and TAP2 subunits in live cells. PCK was incorporated efficiently at K511. Using TAP2-deficient human cells, the authors demonstrated successful light-induced transport of a fluorescently-labeled peptide into the endoplasmic reticulum with this system. Next, they observed photoinduced MHC I transport to the cell surface. Both TAP and TAP's core subunit alone were caged (the latter is less effi-cient). The approach allowed them to trigger and follow MHC I trafficking with precise control, as a de-fined starting point was generated for this cellular process thereby separating the kinetics of PLC as-sembly from peptide translocation and MHC I trafficking. The authors identified an intriguing 1 h lag phase.

Overall, this approach applies genetic code expansion to the exciting intersection of chemical biology and immunology. The method offers tight temporal control over antigen presentation, and, unlike other methods, no cell-impermeable inhibitors need to be delivered, and antigen presentation can be activat-ed in a synchronized fashion, with both loss- and gain-of-function in the same context. However, in its present form, the manuscript does not follow a peptide all the way from the cytosol to the cell surface. Tracing the full course of the pathway through imaging would add further to this impactful manuscript. This research is in a high-impact area and was conducted with rigorous experimental design, and it presents a valuable, generalizable technique for control of antigen presentation (and other processes controlled by P-loop ATP/GTPases) that can be used in live cells. Thus, I strongly recommend publi-cation in Nature Communications Biology after the following, relatively minor revisions:

1) PCK concentration: the use of 20 nM PCK for efficient protein expression is unprecedented. PCK was used at 1 mM when first reported (with the same synthetase), and most investigators use 0.1 to 1 mM concentrations of UAAs. A reported attempt to go down to 10 μ M UAA showed negligible incorporation at this low concentration (see [dx.doi.org/10.1002/anie.201108231](https://doi.org/10.1002/anie.201108231)), and having success 500 times below this concentration is surprising. Please make sure that nano-molar (nM) is correct, and μ M concentrations were not used. If nM is correct, further testing in Figure S2 may be warranted to go above 0.5 μ M PCK and see if expression is fully saturated at that concentration.

Also, further discussion of this surprising result is warranted.

2) To demonstrate complete control over antigen presentation and to increase the impact of this work, the fluorescently-labeled NSTAF647 peptide (or another substrate) should be followed throughout the entire pathway by fluorescence microscopy. If the system presented here is fully functional, when cells are washed and allowed to recover after permeabilization and irradiation, the peptide should be visible on the cell surface and should colocalize with MHC I. The fate of the NSTAF647 peptide should also be clarified in the results section. If it is not translocated to the ER, does it diffuse out of permeabilized cells and get washed away? If the cells are resealed (e.g., using Ca²⁺), is it trapped in the cytosol?

3) It would be important to know if MHC I levels on the cell surface continue to increase after 4 h post-illumination by running the assay shown in Figure 5 until it plateaus out. This saturation should be achieved, as it will showcase the maximum antigen presentation accessible with this methodology and address the author's stated concern about reduced expression levels and the potential for protein turnover to reduce activity.

4) The reason why forward scatter is reduced on cell permeabilization should be given in order to explain Figure S3 to the unfamiliar reader. Aren't flow cytometry readouts using fluorescent proteins more common for assessing permeabilization? Further, based on previous work by the authors (doi.org/10.1038/ncomms7199), can the 10 ng/μL concentration of Streptolysin O permeabilize the ER membrane in addition to the plasma membrane, resulting in a baseline level of nonspecific peptide entry?

5) In Figure S1, is APC-Fire750-labeled anti-HLA antibody displayed on the y-axis? If so, please clarify it by mentioning it in the figure caption and explain why no rescue of MHC I expression is detected on the cell surface when coreTAP or TAP (without amber mutations) are expressed. If not, a histogram of mVenus fluorescence rather than a dot plot is more suitable.

6) The graph in Figure S4 is a little difficult to follow. Each bar could be labeled with a – or + sign for hv, PCK, Bock, TAG mutation, and ATP or ADP. Further, the table of multiple comparisons could be removed and replaced by adding brackets and asterisks for the three that reached significance directly on the graph.

7) Methods:

a. Synthesis: please add a general synthetic methods section specifying the manufacturer of instrumentation (NMR, HRMS) and the C18 silica gel (as well as the type). Specify that the synthesis is modified from Gautier et al. and provide the citation again. Synthetic yields should be added to each step in Supplementary Scheme 1. The ionization method used for mass spectrometry should be specified.

b. Expression constructs: please provide primer sequences for cloning and mutagenesis, and maps of new plasmids.

c. Please clarify if the UAAs were supplemented starting at the time of transfection or starting 5.5 h after transfection when the medium was exchanged.

d. Immunoblotting: antibody dilution factors should be provided, and the type/manufacturer of secondary antibodies should be specified.

e. MHC I surface presentation: the rationale for staining with biotinylated anti-HLA antibodies before incubating and later staining with APC-Fire750-labeled anti-HLA antibodies should be stated.

Is this meant to mask any MHC I remaining after the acid treatment?

f. Peptide transport assay: please specify how the peptide was obtained.

8) To illustrate how PCK decaging works, the full UAA structure should be shown in Figure 1b in order to increase accessibility for the casual reader.

9) A loading control should be provided for Figure 3a to show that the same amount of protein is loaded in each lane. It would then be appropriate to normalize the in-gel fluorescence signal to the loading control in each lane. Transferring the lower portion of the gel and immunoblotting for actin, tubulin, etc., control blots may be suitable.

10) It should be noted for the discussion of protein half-life that TAP typically shows “slow turnover and long half-life” ([dx.doi.org/10.1016/s1074-7613\(00\)80474-4](https://doi.org/10.1016/s1074-7613(00)80474-4)), minimizing the potential limitation identified in the discussion about the use of this approach in a long experiment.

11) To introduce the context of this work, other reports in which photocaged lysines were used to control ATP binding sites would be worthwhile to cite in the introduction or discussion section, including kinases <https://doi.org/10.1002/cbic.201900757> and <https://www.jbc.org/content/295/25/8494.long>) and the helicase UvrD (<https://doi.org/10.1002/cbic.201600624>).

Point-to-point Response:

Reviewer #1 (Kajsa M. Paulsson, Lund University – Expert in Antigen Presentation):

The authors have established a light-controlled system to study the part of the antigen processing chain starting with transport of peptides by TAP. The TAP peptide transport is essential for efficient loading of peptides onto MHC class I molecules and after quality control of the resulting peptide-MHC-class I complexes presentation of the peptides to the immune defense cells on the cell surface. This paper shows that the method using an unnatural amino acid: photo-caged nitrioperonyl caged lysine (PKC) in the catalytic ATP-binding site of TAP allows control of the peptide transport step taking peptides from the cytosol to the ER lumen and the waiting MHC class I molecules. The method presented in the paper provides a very useful tool that if successfully applied can be used to widely study PLC assembly separated from peptide translocation and MHC class I transport.

This paper is a nice continuation from previous work, including the work done where TAP was arrested and re-activated through a photo-conditional variant of the viral inhibitor ICP47. The method presented will be of importance for immunologists and others wanting to advance the understanding of antigen presentation on a mechanistic level as well as in both health and disease. Many viruses and tumors target components of the antigen processing machinery, including the TAP proteins per se, and the consequences are of relevance not only for understanding the mechanisms and principles but also for finding and developing approaches for immunotherapy and selection of antigenic peptide candidates.

The data are collected and analyzed with a well-planned and careful approach. However, since the type of HLA-I alleles vary in dependence on both the quality control system and the TAP transported peptides it would be appropriate and relevant if the HLA-I alleles of the model cell lines was written out in the material and methods. If not known, the cell lines should be typed and the information provided.

Reply: We thank the reviewer for her insightful and encouraging comments, highlighting the importance and significance of the approach and experimental data, also underlining the advancement using such approach to understand the mechanistic basis of antigen processing.

Action: The HLA-I allomorphs of the model cell line have now been added in the revised manuscript.

The use of statistics seems appropriate but for the reader it could be more clearly explained e.g., what does n refer to and how many times were the experiments repeated with similar outcome. Fig. 3. Was the FACS done in quadruplicates or is the panel to the right four sets pooled from different experiments? Fig. 5. is n, the number of experiments performed at different occasions? Or is n the number of replicates in one experiment?

Reply: We thank the reviewer for bringing up this point. We added further information on the replicates in the revised manuscript.

The conclusions are well-balanced and justified by the data presented in the paper.

Reply: We are delighted about this very positive feedback.

Suggested improvements: experiments, data for possible revision:

The paper is solid and interesting. No further experiments or additional data analysis needed, if the manuscript is complemented with information about the HLA-I alleles in the cell lines.

The references are appropriate and credit previous work in the field.

The abstract could perhaps be improved by emphasis on the functionality of the assay and thereby also justify the choice and development of method.

Reply: Thanks again for the very positive feedback on our work. We modified the abstract accordingly including an emphasis on the functionality of the assay.

In the discussion it would be interesting if the authors related to if it plausible that there are mechanisms for compensation, i.e., increased egress rate or similar that operate when MHC class I molecules are washed away from the cell surface? Additional studies on allelic differences would be of importance and not least, on combination of different HLA-I haplotypes.

Reply: Thanks for this suggestion. We already considered such experiments. We are presently developing mono-allomorphic cells to investigate the allelic difference upon photo-triggered induction of antigen processing. However, these studies are very time-consuming in particular in the COVID-19 situation.

Reviewer #2 (Expert in photo regulation of cellular processes):

The manuscript "Light control of the peptide-loading complex synchronizes antigen translocation and MHC I trafficking" by Brunnberg et al. reports a novel method for triggering the activity of Transporter associated with Antigen Processing (TAP) with light. TAP transports cytosolic peptides into the endoplasmic reticulum so that Major Histocompatibility Complex I (MHC I) can display them on the cell surface, activating the immune system. To attain control over TAP, the authors used genetic code expansion to mutate residue K511 in TAP to an unnatural amino acid, the photocaged lysine PCK. This residue comprises lysine in which the side chain amine is masked with a photocleavable 6-nitropiperonyl protecting group via a carbamate linkage. K511 is essential for TAP to bind to and hydrolyze ATP. Thus, the TAP-K511PCK mutant is inactive until it is converted back the wild-type protein using UV light.

The authors devised a system using a ribosome-skipping site to produce stoichiometric quantities of the TAP1 and TAP2 subunits in live cells. PCK was incorporated efficiently at K511. Using TAP2-deficient human cells, the authors demonstrated successful light-induced transport of a fluorescently-labeled peptide into the endoplasmic reticulum with this system. Next, they observed photoinduced MHC I transport to the cell surface. Both TAP and TAP's core subunit alone were caged (the latter is less efficient). The approach allowed them to trigger and follow MHC I trafficking with precise control, as a de-fined starting point was generated for this cellular process thereby separating the kinetics of PLC assembly from peptide translocation and MHC I trafficking. The authors identified an intriguing 1 h lag phase.

Overall, this approach applies genetic code expansion to the exciting intersection of chemical biology and immunology. The method offers tight temporal control over antigen presentation, and, unlike other methods, no cell-impermeable inhibitors need to be delivered, and antigen presentation can be activated in a synchronized fashion, with both loss- and gain-of-function in the same context. However, in its present form, the manuscript does not follow a peptide all the way from the cytosol to the cell surface. Tracing the full course of the pathway through imaging would add further to this impactful manuscript. This research is in a high-impact area and was conducted with rigorous experimental design, and it presents a valuable, generalizable technique for control of antigen presentation (and other processes controlled by P-loop ATP/GTPases) that can be used in live cells. Thus, I strongly recommend publication in Nature Communications Biology after the following, relatively minor revisions:

Reply: We thank the reviewer for his/her enthusiastic and very insightful feedback on the manuscript, highlighting the exciting intersection of antigen processing, including future approaches to trace the journey of an antigen-MHC I complex within living cells.

1) PCK concentration: the use of 20 nM PCK for efficient protein expression is unprecedented. PCK was used at 1 mM when first reported (with the same synthetase), and most investigators use 0.1 to 1

mM concentrations of UAAs. A reported attempt to go down to 10 μM UAA showed negligible incorporation at this low concentration (see [dx.doi.org/10.1002/anie.201108231](https://doi.org/10.1002/anie.201108231)), and having success 500 times below this concentration is surprising. Please make sure that nano-molar (nM) is correct, and μM concentrations were not used. If nM is correct, further testing in Figure S2 may be warranted to go above 0.5 μM PCK and see if expression is fully saturated at that concentration. Also, further discussion of this surprising result is warranted.

Reply: The reviewer is right that this is an unprecedented and surprising result. We carefully checked the PCK concentration to validate this result. We can confirm that the applied 20 nM of PCK were sufficient for reliable incorporation. We also checked higher concentrations (>0.5 μM) for protein expression to warrant saturation. Also, in this case, no significant increase in the mean fluorescence signal was detected during flow cytometry, indicating that saturation was already reached.

The efficient incorporation is most likely a result of the mutated tRNA synthase (PylRS from *M. mazei*) in combination with the amber stop mutation of Walker A lysine in TAP2. We would like to point out that efficient PCK incorporation was monitored via flow cytometry, whereas Plass *et al* determined the UAA incorporation via an automated microscope procedure. A difference between both methods is the number of measured events as well as the threshold adjustment. Flow cytometry averages over several thousands of cells and thus provides an improved statistic. The automated microscope procedure can average over hundreds of cells. Furthermore, the set adjustments during fluorescence imaging can circumvent the detection of very low expressing cells and thus shift or crop the lower limit for efficient PCK incorporation at low concentration.

Action: We added a statement of caution in the main manuscript to indicate the difference to commonly applied UAA concentrations for incorporation.

2) To demonstrate complete control over antigen presentation and to increase the impact of this work, the fluorescently-labeled NSTAF647 peptide (or another substrate) should be followed throughout the entire pathway by fluorescence microscopy. If the system presented here is fully functional, when cells are washed and allowed to recover after permeabilization and irradiation, the peptide should be visible on the cell surface and should colocalize with MHC I. The fate of the NSTAF647 peptide should also be clarified in the results section. If it is not translocated to the ER, does it diffuse out of permeabilized cells and get washed away? If the cells are resealed (e.g., using Ca²⁺), is it trapped in the cytosol?

Reply: We thank the reviewer for this excellent suggestion. On the long term we follow exactly these plans. For this, we were trying to establish mono-allomorphic cell lines, which harbor only one HLA allele (see response to reviewer #1). However, these experiments are very demanding and time-consuming. In addition, the reviewer should note that we are using peptides which harbor an N-core glycosylation sequence that traps translocated peptides in the ER lumen. These peptides provide an

excellent readout; however, they cannot be loaded on MHC I molecules. In the future, we are investigating in fluorogenic dyes, which can react with peptides loaded on MHC I molecules.

3) It would be important to know if MHC I levels on the cell surface continue to increase after 4 h post-illumination by running the assay shown in Figure 5 until it plateaus out. This saturation should be achieved, as it will showcase the maximum antigen presentation accessible with this methodology and address the author's stated concern about reduced expression levels and the potential for protein turnover to reduce activity.

Reply: We thank the reviewer for his/her suggestion. We tracked the MHC I levels on the cell surface over a time period of 8 h, but found that the increase reaches its maximum 4 h after illumination. After these 4 h the values decreased slightly. Since we did not examine this decrease further, we have decided to show only the increase of MHC I level. As we are working with a system in which TAP can only be activated once and is then degraded after a certain time, a decrease in the MHC I molecules expressed on the cell surface is not surprising.

4) The reason why forward scatter is reduced on cell permeabilization should be given in order to explain Figure S3 to the unfamiliar reader. Aren't flow cytometry readouts using fluorescent proteins more common for assessing permeabilization? Further, based on previous work by the authors (doi.org/10.1038/ncomms7199), can the 10 ng/μL concentration of Streptolysin O permeabilize the ER membrane in addition to the plasma membrane, resulting in a baseline level of nonspecific peptide entry?

Reply: We would like to thank the reviewer for bringing up this point. As shown in Döring *et al.* (DOI: 10.1182/bloodadvances.2018027268), the gating on semi-permeabilized cells by using the forward and side scatter is an established procedure.

Döring *et al.* (2019) *Blood Adv.*, Supplementary Figure 1

The concentration of streptolysin O used for semi-permeabilization is already carefully tested and optimized for the experiment. We try to reach only 80-90% semi-permeabilization, which ensures that the ER remains intact. The compartmentalization is nicely demonstrated by the accumulation of fluorescent peptides in the ER lumen.

5) In Figure S1, is APC-Fire750-labeled anti-HLA antibody displayed on the y-axis? If so, please clarify it by mentioning it in the figure caption and explain why no rescue of MHC I expression is detected on the cell surface when coreTAP or TAP (without amber mutations) are expressed. If not, a histogram of mVenus fluorescence rather than a dot plot is more suitable.

Reply: We thank the reviewer for this comment. Figure S1 displays representative dot plots corresponding to the graph shown in Figure 2b. In these experiments we focused on TAP expression and MHC I was not stained. The APC-Fire750 channel was used to create dot plots of the single cells as this shows a distinct emerging population compared to histograms.

6) The graph in Figure S4 is a little difficult to follow. Each bar could be labeled with a – or + sign for hv, PCK, Bock, TAG mutation, and ATP or ADP. Further, the table of multiple comparisons could be removed and replaced by adding brackets and asterisks for the three that reached significance directly on the graph.

Reply: Thanks for this suggestion. Figure S4 shows a variety of different conditions. In order to be easier to understand, we changed Figure S4 in the revised manuscript.

7) Methods:

a. Synthesis: please add a general synthetic methods section specifying the manufacturer of instrumentation (NMR, HRMS) and the C18 silica gel (as well as the type). Specify that the synthesis is modified from Gautier et al. and provide the citation again. Synthetic yields should be added to each step in Supplementary Scheme 1. The ionization method used for mass spectrometry should be specified.

Reply: All required information has been added to the Online Methods.

b. Expression constructs: please provide primer sequences for cloning and mutagenesis, and maps of new plasmids.

Reply: All information has been added to the data source file.

c. Please clarify if the UAAs were supplemented starting at the time of transfection or starting 5.5 h after transfection when the medium was exchanged.

Reply: UAAs were added along with the medium change 5.5 h after transfection. We added this information in the revised manuscript accordingly.

d. Immunoblotting: antibody dilution factors should be provided, and the type/manufacturer of secondary antibodies should be specified.

Reply: We added the antibody dilution factors for immunoblotting in the revised manuscript accordingly.

e. MHC I surface presentation: the rationale for staining with biotinylated anti-HLA antibodies before incubating and later staining with APC-Fire750-labeled anti-HLA antibodies should be stated. Is this meant to mask any MHC I remaining after the acid treatment?

Reply: To mask any remaining peptide-MHC I complexes and to block the cell surface, we stained the cells with a biotinylated anti-HLA antibody immediately after the acid treatment. As reported in Döring et al. (DOI: 10.1182/bloodadvances.2018027268), newly surfaced MHC I will not be stained by the biotinylated antibody and can be quantified by the APC-Fire750-labeled anti-HLA antibody. This has been clarified in the revised manuscript and additional information was added accordingly.

f. Peptide transport assay: please specify how the peptide was obtained.

Reply: This information has been added accordingly.

8) To illustrate how PCK decaging works, the full UAA structure should be shown in Figure 1b in order to increase accessibility for the casual reader.

Reply: Thanks for this helpful suggestion. Figure 1b is now changed accordingly.

9) A loading control should be provided for Figure 3a to show that the same amount of protein is loaded in each lane. It would then be appropriate to normalize the in-gel fluorescence signal to the

loading control in each lane. Transferring the lower portion of the gel and immunoblotting for actin, tubulin, etc., control blots may be suitable.

Reply: When loading, we made sure to load the same number of cells in each lane. This is also described in detail in the Online Methods under "In gel fluorescence and immunoblotting". Figure 2 also shows that a constant amount of actin was detected when loading the same number of cells.

10) It should be noted for the discussion of protein half-life that TAP typically shows "slow turnover and long half-life" ([dx.doi.org/10.1016/s1074-7613\(00\)80474-4](https://doi.org/10.1016/s1074-7613(00)80474-4)), minimizing the potential limitation identified in the discussion about the use of this approach in a long experiment.

Reply: The reviewer is right that native TAP shows typically a long half-life. For our approach, we transiently transfected TAP2-deficient cells with amber-suppressed TAP2 mutants. Thus, TAP turnover could be affected by the expression system and dilution due to cell division. Further, active TAP is exclusively generated by illumination. After uncaging of PCK, *de novo* TAP2 expression is terminated by the amber stop codon and degradation of light-activated TAP might be relevant. When we tracked light-triggered MHC I presentation over 8 h, we found that MHC I levels slightly decrease after 4 h (see response to reviewer #2, comment 3). This suggests that the half-life of photo-conditional TAP could be decreased compared to native TAP. We added this part of the discussion as a statement of caution for extended experiments or expansions of the photo-conditional approach to other targets.

11) To introduce the context of this work, other reports in which photocaged lysines were used to control ATP binding sites would be worthwhile to cite in the introduction or discussion section, including

kinases <https://doi.org/10.1002/cbic.201900757> and <https://www.jbc.org/content/295/25/8494.full> and the helicase UvrD (<https://doi.org/10.1002/cbic.201600624>).

Reply: We like to thank the reviewers for bringing up this additional reference, which are now included in the revised manuscript. One of the references has already been cited in our initial submission.

REVIEWERS' COMMENTS:

Reviewer #1 (Remarks to the Author):

The authors have in a satisfying way addressed the issues identified in the first draft with appropriate additional experimental work and in writing. I am pleased with the revised version of the manuscript and recommend it for publication. In addition, I look forward to a future paper where mono-allomorphic cells are used to investigate the allelic difference upon photo-triggered induction of antigen processing.

/Kajsa M. Paulsson, Lund University

Reviewer #2 (Remarks to the Author):

The authors sufficiently addressed all of my comments and I would like congratulate them to a wonderful manuscript, which can now be published after the following minor revision:

The structure of the released "photocage" in Fig 1B is incorrect. Please change the nitro group (NO₂) to a nitroso group (NO).